# Time Perspective as a Mediator of Depressive Symptoms in Patients with Polycystic Ovary Syndrome

**DOI:** 10.3390/healthcare11070993

**Published:** 2023-03-30

**Authors:** Agnieszka Adamczak, Włodzimierz Płotek, Aleksandra Głowińska, Małgorzata Sobol, Ewa Wysocka, Grzegorz Polak, Izabela Dymanowska-Dyjak, Julia Spaczyńska, Łukasz Adamczak, Beata Banaszewska

**Affiliations:** 1Department of Laboratory Diagnostics, Poznań University of Medical Sciences, 60-569 Poznań, Poland; 2Department of Anaesthesiology and Intensive Care, Medical University of Lublin, 20-954 Lublin, Poland; 3Faculty of Psychology, University Hospital of Obstetrics and Gynaecology, 60-535 Poznań, Poland; 4Department of Psychology, University of Warsaw, 00-927 Warsaw, Poland; 5Independent Laboratory of Minimally Invasive Gynecology and Gynecological Endocrinology, Medical University of Lublin, 20-059 Lublin, Poland; 6UCL Medical School, London WC1E 6DE, UK; 7Department of Reproduction, Poznań University of Medical Sciences, 60-535 Poznań, Poland

**Keywords:** polycystic ovary syndrome, depressive symptoms, time perspective

## Abstract

Background: Polycystic ovary syndrome (PCOS) is a chronic endocrinopathy characterized by oligo- or anovulation, clinical and/or biochemical markers of hyperandrogenism, and polycystic ovaries, and it is associated with an increased prevalence of depression. Research conducted on psychiatric patients has shown correlations between depression and decreased cognitive function. The aim of this study was to examine the possible mediation of the time perspective (TP) in the development of depressive symptoms in patients with PCOS. Methods: A study was conducted on 83 patients with PCOS and 65 healthy women. Standardized questionnaires were used to assess depressive symptoms (Beck Depression Inventory—BDI-II) and time perspective (Zimbardo Time Perspective Inventory—ZTPI). Results: Our study revealed an indirect influence of depressive symptoms on PCOS through the positive future time perspective. In the logistic regression model, which included depression and a given time perspective as predictors of PCOS, only the future TP (β = −0.004, *p* < 0.003, OR = 1.004, 95% CI [1.001, 1.008]) was significantly independently related to the occurrence of PCOS. Conclusions: Our result is another argument for the role of psychoeducation and appropriate communication with a patient from the risk group in a way that builds hope and allows to regain influence on life situation.

## 1. Introduction

Polycystic ovary syndrome (PCOS) is an endocrine disorder diagnosed in approximately 8–10% of women of reproductive age, and it is one of the most common endocrinopathies. To diagnose PCOS in accordance with the Rotterdam criteria, at least two of the three following criteria must be present: oligo- or anovulation, clinical and/or biochemical markers of hyperandrogenism, and polycystic ovaries (identified by ultrasound) [1,2]. Women with PCOS are more likely to develop infertility, obesity, and cardiovascular diseases such as dyslipidemia [3], type 2 diabetes [4], atherosclerosis, and hypertension [5,6]. The pathogenesis of PCOS is complex and involves contributions of environmental, genetic, and transgenerational components.

PCOS causes complex hormonal changes affecting the function of the central nervous system [7]. Its two basic aspects are emotions and cognitive processes. Researchers have noted significantly elevated rates of depression and anxiety in PCOS patients. Studies based on psychological tests have shown the decline of cognitive functions, especially language functions, in PCOS patients [8]. Researchers have observed that insulin and sex hormones such as androgens influence cognitive processing [9,10]. Research conducted on psychiatric patients has shown a correlation between depression and decreased cognitive function [11,12]. PCOS is associated with various psychological determinants and mental health problems such as depression, personality disorders, eating disorders, body dissatisfaction, anxiety, diminished sexual satisfaction, and lowered health-related quality of life [13,14,15]. There have been multiple reports showing high depression scores among women with PCOS [13,16,17]. For many years, there have been some scientific publications whose authors linked PCOS with an increased risk of mood disorders [18]. The literature suggests that there is an overlap of clinical symptoms between depression and PCOS, and therefore, there is a possibility of common associations between depression, PCOS, and PCOS-associated abnormalities [19,20] Although the mechanism of the higher prevalence of depression in PCOS patients is not clear, factors such as high BMI, infertility, hyperandrogenemia/hyperandrogenism, and insulin resistance may play collective roles in its development [21]. It has been hypothesized that depression in PCOS may be caused by hormonal disorders including dysregulation of the hypothalamic-pituitary-adrenal (HPA) axis. The HPA disorders hypothesis assumes abnormal secretion of GnRH in the hypothalamus, which increases serum LH concentrations. It is suspected that the increased sensitivity of the pituitary gland to GnRH or incorrect, too-frequent generation of GnRH pulses caused by a decrease in the secretion of neurotransmitters that inhibit the activity of the hypothalamus (e.g., dopamine or hormones from the endorphin group) play an important role in the described mechanism [22,23]. Some researchers have suggested there might be a two-way relationship between inflammation and depression in this population [24]. PCOS is a state of chronic inflammation, and there have been studies describing the relationship between PCOS and some inflammatory markers [25].

Time perspective (TP) is a term developed by Zimbardo. TP measures one’s positive and negative attitude to the past and future and hedonistic and fatalistic attitude to the present. It refers to the human concept of integration and evaluation of one’s temporal position and evaluation of emotions, memories, and experiences, thus linking various aspects of emotions and cognition. For example, a negative view of the past could develop a pessimistic and depressive attitude to the present or the future. Researchers have observed that the alteration of time dimensions is often related to depressive disorders [26,27]. Thus far, there have been no scientific studies evaluating TP expression in PCOS patients, especially in the context of depressive symptoms.

Therefore, the aim of this study was to examine the possible mediation of TP in the development of depressive symptoms in patients with PCOS. The following research hypothesis was assumed: there is a positive relationship between PCOS and depressive symptoms, and this relationship is explained by TP. The authors of this research assumed that TP is related to psychosomatic symptoms of PCOS because it is significantly related to the quality of life, stress, and health behavior [28,29].

## 2. Materials and Methods

### 2.1. Study Group

One hundred and sixty-three women of reproductive age were recruited in the study. The study was conducted on 148 women (mean age 28.6 years) who were divided into two groups. All patients admitted to the Department of Infertility and Reproductive Endocrinology were invited to participate in the study. Five patients refused to participate in it. Ten patients were excluded from participation in the study because they were taking psychiatric drugs—antidepressants. We recruited women from the beginning of October 2021 to the end of March 2022. One group (83 women, mean age 26.8 years) included patients with PCOS diagnosed according to the Rotterdam criteria [30]. The other group (65 women, mean age 30.8 years) included patients without PCOS. Polycystic ovaries were diagnosed in accordance with the latest recommendations/guidelines if the ovary had 20 subcapsular antral follicles with a diameter of 2–9 mm, or the volume of the ovary was at least 10 mL [2].

### 2.2. Inclusion Criteria

All the patients needed to meet the following inclusion criteria to be enrolled in the study: written consent to participate in the study, Polish nationality, and 18–45 years of age. The PCOS group also had to meet two of the three PCOS criteria: oligo- or amenorrhea, hyperandrogenism (hirsutism or acne)/hyperandrogenemia (increased total serum testosterone level > 0.6 ng/mL), and polycystic ovaries.

### 2.3. Exlusion Criteria

The women who met one or more of the following exclusion criteria were not qualified for the study: refusal to participate in the study, age over 45 years, nationality other than Polish, psychiatric disorder, taking psychiatric drugs, Mini-Mental State Examination (MMSE) score under 27 points, and evidence of other endocrinopathies such as congenital adrenal hyperplasia, Cushing syndrome, and androgen-secreting tumors.

The patient-selection process is shown in Figure 1.

### 2.4. Biochemical Research

The serum testosterone levels were measured with specific chemiluminescence assays (Chiron Diagnostics GmbH, Fernwald, Germany). All the women had their venous blood samples collected during the early follicular phase. The samples were collected between 7 and 8 a.m. and immediately analyzed in a certified central laboratory at the Obstetric and Gynecologic University Hospital in Poznań. The sampled sera were stored at −20 °C for subsequent analysis.

### 2.5. Psychological Research

The study used a diagnostic survey with questionnaires. The following instruments were applied: Mini–Mental State Examination (MMSE), Beck Depression Inventory (BDI-II), and Zimbardo Time Perspective Inventory (ZTPI).

The baseline cognitive impairment was determined using the MMSE [30]. This screening scale consists of 30 questions/tasks enabling the quantitative assessment of various aspects of cognitive functioning. The areas assessed included time orientation, spot orientation, memorization, attention and counting, reminding, naming, repetition, understanding, reading, writing, and drawing. Due to the significant impact of age and education on the result, it was proposed that scores under 27 points should be used as the basis for further detailed clinical examination to confirm or exclude dementia. Cronbach’s alpha coefficient for the entire normalization sample was 0.82 [30].

Depression was measured with BDI-II. This questionnaire contains 21 items. Each item is rated on a 4-point scale ranging from 0 to 3. The lowest score is 0, whereas the highest is 63 points. Depression is suspected if the score is ≥14 points. Cronbach’s alpha coefficient for the entire normalization sample was 0.92 [31]. The Polish adaptation of the BDI-II was used in our study [31].

Time perspective was assessed with ZTPI [32]. This questionnaire of time perception is the Polish adaptation and consists of 20 items. The positive-past scale measures the concentration on the positive past. The negative-past scale measures the concentration on the negative past, i.e., thinking about unpleasant memories and failures. The hedonistic-present scale refers to the focus on momentary pleasures regardless of the consequences of one’s impulsive actions. The future scale measures the focus on the future, i.e., focusing on planning, success, and consistent implementation of one’s goals in life. Cronbach’s alpha coefficient for the entire normalization sample was 0.8 [32].

The study was approved by the Institutional Ethical Committee of the Poznań University of Medical Sciences (742/20). The research was voluntary and anonymous, and the results were used for scientific purposes.

### 2.6. Statistical Analysis

The following tools were used for statistical analyses: Statistica (data analysis software system) version 13 (TIBCO Software Inc., Tulsa, OK, USA, 2017) and R: A language and environment for statistical computing (R Core Team, Vienna, Austria, 2022) version 4.1.3 (R Foundation for Statistical Computing, Vienna, Austria). The quantitative variables were presented as the mean and standard deviation. The qualitative variables were presented as quantity and percentage. The Shapiro–Wilk test was used to the normality of data distribution. The homogeneity of variance was checked with the Levene’s test. Student’s parametric *t*-test for independent samples was used to check the difference between two quantitative variables if the data fitted normal distribution. The non-parametric Mann–Whitney test was used to compare non-normally distributed data. Pearson’s chi-square test was used to examine differences between the qualitative variables. Logistic regression and multiple regression were used to analyze some data. In the first step of the analysis, two regression models were tested for each subscale of the ZTPI with CMA (causal mediation analysis) to investigate whether the relationship between depression and PCOS is mediated by the time perspective. The *p*-value of less than 0.05 was considered significant. The Power Calculator—the *t*-test for two groups—was used to analyze the results, the samples’ sizes, and the probability of type 1 error. Values above 0.8 indicate high test power. Analysis of the statistical power comparison of future TP between the study group and the control group showed that the power of the test was 1.

## 3. Results

Table 1 shows the characteristics of the groups. There were 148 patients with a mean age of 28.6 (18.0–40.0) years. The mean BMI was 26.4 (16.9–45.9) kg/m^2^ (PCOS patients—27.7 kg/m^2^, healthy subjects—24.7 kg/m^2^). The groups differed significantly in their age (*p* < 0.001) and BMI (*p* = 0.001). The women with the PCOS were younger and had a statistically higher BMI. The groups did not differ significantly in their education. However, there was a significant difference in depressive symptoms (*p* < 0.001) between the groups. The PCOS patients had significantly more depressive symptoms that the healthy women. The MMSE did not reveal statistically significant differences between the groups. The groups differed significantly in the future time perspective (*p* < 0.001). There was not a significant difference between other time perspectives.

For each time perspective, in the first least squares regression model, depression was significantly associated with the intensity of the positive-past TP (β = −0.19, 95% CI [−0.25, −0.13], t (145) = −6.21, *p* < 0.001), negative-past TP (β = 0.18, 95% CI [0.11, 0.26], t (145) = 4.80, *p* < 0.001), and future TP (β = −0.11, 95% CI [−0.17, −0.05], t (145) = −3.58, *p* < 0.001). However, there was no relationship between depression and the intensity of the hedonistic TP in the group under study (β = 0.00, 95% CI [−0.07, 0.07], t (145) = 0.06, *p* = 0.951).

In the second logistic regression model, which included depression and a given time perspective as predictors of PCOS, only the future TP (β = −0.004, *p* < 0.003, OR = 1.004, 95% CI [1.001, 1.008]) was significantly independently related with the occurrence of PCOS (Table 2). As the future-looking perspective decreased, and the level of depression was controlled, the chance of having PCOS increased by 0.4% for each unit increase. The increase in depression was negatively related to the intensity of future orientation. The stronger the symptoms of depression, the lower the orientation towards the positive future and, at the same time, the higher the probability of PCOS.

The analysis of all temporal perspectives is burdened with the lack of control of the relationships between individual perspectives. The pairwise correlation analysis showed that individual perspectives were related to each other (r_pn_ = −0.39, *p* < 0.001; r_pf_ = 0.35, *p* < 0.001; r_ph_ = 0.05, *p* = 0.52; r_nf_ = −0.15, *p* = 0.07; r_nh_ = 0.21, *p* = 0.01; r_fh_ = 0.06, *p* = 0.44, where *p*—positive-past TP; n—negative-past TP; f—future TP; h—hedonistic TP). Therefore, the mediating role of time perspective in the influence of depression on PCOS was investigated by controlling the remaining variables with structural equations (with the R program, the lavaan package). The data collected in the study were analyzed by means of structural equations with the DWLS method and robust calculation of the measurement error based on a bootstrap of 5000 samples. The analysis showed the “pure” effect of individual variables, i.e., the impact of each variable on the probability of PCOS occurrence assuming control of all the rest. The final path model is shown in Figure 2. The values in the chart represent the non-standardized beta. The causal mediation analysis (CMA) showed that only the future variable mediated the influence of depression on the occurrence of PCOS. As the level of depression increased, the future prospects decreased, and this decrease increased the likelihood of PCOS.

## 4. Discussion

The association between PCOS and depressive symptoms has already been well studied. However, there are still numerous questions about the mechanism of this relationship. Although the role of biochemical factors seems obvious, it is difficult to capture specific relationships in research. Nevertheless, the role of psychosocial factors is apparent. The influence of TP on the quality of life and the role of TP in the mechanisms of mood disorders were observed independently [29,33]. Therefore, it was important to study TP on women with PCOS. To our knowledge, there has never been any research on this subject before.

The aim of our study was to investigate the role of time perspective in the relationship between depressive symptoms and PCOS. This result should be analyzed with caution; however, it gives us some idea of prevention and further research. This result is another argument for the role of psychoeducation and appropriate communication with a patient from the risk group in a way that builds hope and allow to regain influence on life situation. At the time of diagnosis, working with an interdisciplinary team including a psychologist may help patients’ TP to remain optimistic, and this could be a preventive factor for depression associated with PCOS. Beyond the multifactorial physiopathology of PCOS and depression, the diagnosis can also be complex [34,35]. Therefore, studies of the relationship between them causes many difficulties for scientists. Many studies indicate that it is difficult to expect a breakthrough discovery. Thus, it is left to scientists to study these abnormalities piece by piece. The obtained result is believed to be just such an element, and it may be a valuable clue for use in psychoeducational programs. A positive relationship was found between depressive symptoms and PCOS, and this relationship was explained by time perspective. The tendency to focus on the positive future was a mediator of the link between depressive symptoms and PCOS. These findings can be explained by the fact that the positive future time perspective affects young women’s thinking about their plans and dreams, including those about starting a family and having children. Depressive symptoms are associated with future anxiety, avoidance of thinking about plans and dreams, and negative perception of the future [29]. It is probable that the attitude towards the future increases the psychosomatic symptoms strongly related to female attractiveness, fertility, and the possibility of finding a partner. The manifestations of PCOS, including acne, hirsutism, androgenic alopecia, obesity, and other symptoms, lead to a loss of female characteristics and make the patients feel less attractive, which negatively affects their mental health due to lower self-esteem and physical satisfaction [16,36]. The role of positive thinking in the treatment of these symptoms of PCOS is emphasized in scientific publications [37,38]. The relationships between many aspects of psychological health and PCOS have not yet been fully investigated. Several studies have suggested that depression and PCOS are inflammatory disorders characterized by increased levels of inflammatory markers. There has been growing evidence that depressive symptoms are associated with a systemic immune activation, including abnormality in immune cell numbers and inflammatory markers [39]. Multiple meta-analyses [40,41] have established that individuals with depressive symptoms have elevated levels of interleukin-6 (IL-6), tumor necrosis factor (TNF-α), and C-reactive protein (CRP) in the blood compared to healthy controls [42,43]. Correlations have been also found between increased levels of these markers in PCOS women compared with healthy women [44,45]. It would appear that there might be an inflammatory relationship between depression and PCOS [24,25]. The results of our study may be the basis for deepening the theory assuming the non-specificity of the psychosomatic mechanism of chronic diseases, including PCOS. In particular, this applies to TP as a factor that increases susceptibility to stress, which in combination with the genetic background might be a factor of PCOS [17,46].

In our analysis, we observed that the women with PCOS had higher BMI than the healthy subjects. Over many years, there has been accumulating evidence that overweight is more common in PCOS patients [47,48]. A higher BMI is another factor that contributes to depression. Researchers often emphasize the relationship between the occurrence of depression or lower quality of life and obesity [49,50]. The relationship between these two conditions is bidirectional: the occurrence of one increases the risk of the other [51].

Our study showed that the women with PCOS had significantly more depressive symptoms than the healthy women. Our results were in line with the findings of the research conducted by Damone et al. and Farell et al. [17,52]. In our study, depressive symptoms were detected with a highly evaluated screening tool that can be easily used by every doctor without psychological supervision. This broadens the spectrum of diagnostic tools for clinicians in gynecology. The BDI can be easily applied and interpreted in everyday gynecologic practice. Due to the high occurrence of depression in the population under study, a BDI test should be considered to every patient with PCOS.

The MMSE, which was applied as a screening tool for preexisting cognitive deficit, enabled the creation of more homogenous groups. There was no statistical difference between the women in both groups. The patients in our groups had a similar level of education, which was crucial to ensure that the education did not act as a confounding factor.

There has not been any research on this subject before. Our study raised many questions in need of further investigation, which will enable comprehensive characterization and targeting of the biological and psychological links between PCOS and depression.

Our study was limited by the following factors: the number of participants, questionnaire-based and self-report data, and cross-sectional design. It resulted only in statistical prediction, and conclusions should only be made very cautiously. The women with PCOS were younger than the healthy subjects because the PCOS patients experienced disturbing symptoms relatively early and sought diagnosis. They were hospitalized earlier and underwent biochemical and hormonal tests.

## 5. Conclusions

Our study revealed an indirect influence of depressive symptoms on PCOS through the positive future time perspective. Better ability to focus on the positive future should be emphasized in therapeutic interventions applied to PCOS patients. Our findings highlight the significant role of the biopsychosocial perspective in the treatment of PCOS by a multidisciplinary team. This is the first study assessing TP expression in PCOS patients, especially in the context of depressive symptoms. This is a practical issue for the effectiveness of psychotherapy, and it should be taken into consideration in future studies conducted on a larger number of PCOS patients. The results still leave a number of doubts and require further studies. Our article remains open. We trust that further research inspired by the results described here will bring new, interesting results, leading to valuable practical aspects.

## Figures and Tables

**Figure 1 healthcare-11-00993-f001:**
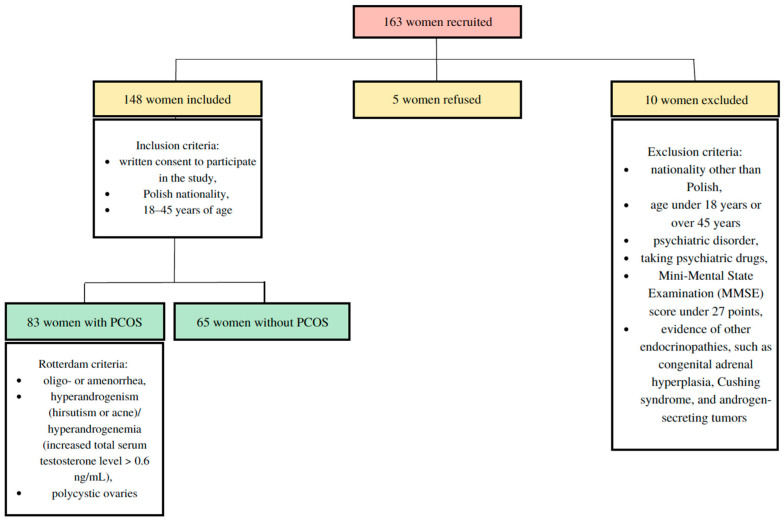
Patient-selection process.

**Figure 2 healthcare-11-00993-f002:**
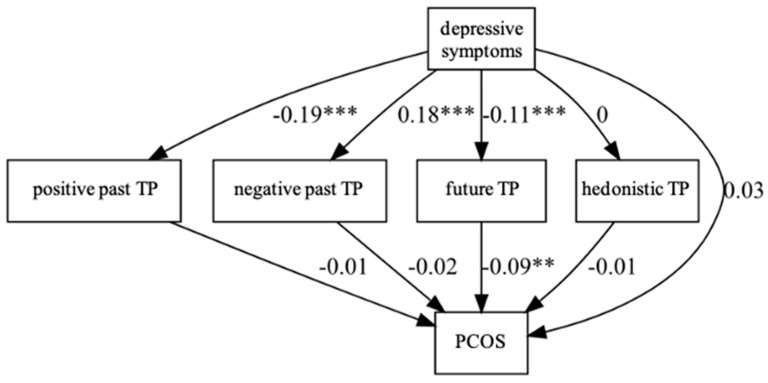
The relationships between PCOS and depressive symptoms and time perspective. ** *p* < 0.01; *** *p* < 0.001.

**Table 1 healthcare-11-00993-t001:** Group characteristics.

Parameters		All Women n = 148	PCOS n = 83	Without PCOS n = 65	*p*-Values
			Mean ± SD/% (n)		
Mean age (years)		28.6 ± 5.2	26.8 ± 4.8	30.8 ± 4.9	<0.001 ^a^
BMI (kg/m^2^)		26.4 ± 5.8	27.3 ± 5.8	23.2 ± 5.3	0.001 ^b^
Education	Primary	8 (12)	4 (6)	4 (6)	0.843 ^c^
	Secondary	31 (46)	17 (25)	14 (21)	
	Higher	61 (90)	35 (52)	26 (38)	
Depressive symptoms		28 (42)	42 (35)	11 (7)	<0.001 ^c^
MMSE results		29.3 ± 0.8	29.2 ± 0.8	29.4 ± 0.8	0.214
ZTPI	positive past TP	19.6 ± 3.9	18.9 ± 0.4	20.1 ± 0.5	0.076 ^a^
	negative past TP	15.8 ± 4.7	16.0 ± 0.6	15.4 ± 0.5	0.481 ^a^
	future TP	19.4 ± 3.6	18.5 ± 0.4	20.5 ± 0.4	<0.001 ^a^
	hedonistic TP	14.3 ± 4.1	14.2 ± 0.5	14.4 ± 0.5	0.764 ^a^

Mean, average value; SD, standard deviation; BMI, body mass index; PCOS, polycystic ovarian syndrome; BDI-II, Beck Depression Inventory; MMSE, Mini-Mental State Examination; ZTPI, Zimbardo Time Perspective Inventory; TP, time perspective. ^a^ Student’s *t*-test; ^b^ Mann–Whitney test; ^c^ Pearson’s chi-square test.

**Table 2 healthcare-11-00993-t002:** The coefficients for each time perspective.

Mediator	Effect	Beta (β)	*p*-Value	OR	CI.L	CI.U	%
Positive-past TP	ACME	−0.001	0.712	0.999	0.994	1.004	5.3
	ADE	0.015	0.004	1.015	1.004	1.027	
	Total	0.014	0.004	1.015	1.004	1.026	
Negative-past TP	ACME	−0.001	0.472	0.999	0.995	1.002	9.4
	ADE	0.015	0.004	1.015	1.004	1.027	
	Total	0.014	0.013	1.014	1.003	1.026	
Future TP	ACME	−0.004	0.003	1.004	1.001	1.008	20.4
	ADE	0.015	0.005	1.015	1.004	1.025	
	Total	0.018	0.000	1.018	1.008	1.029	
Hedonistic TP	ACME	0.000	0.998	1.000	0.999	1.001	0.0
	ADE	0.015	0.007	1.015	1.004	1.027	
	Total	0.015	0.007	1.015	1.004	1.027	

ACME, averaged causal mediated effect; ADE, averaged direct effect; TP, time perspective; OR, odds ratio; CI.L, confidence interval lower; CI.U, confidence interval upper.

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
