# Peer review of "Time Perspective as a Mediator of Depressive Symptoms in Patients with Polycystic Ovary Syndrome"

_healthcare, 2023, doi:10.3390/healthcare11070993_

Round 1

Reviewer 1 Report

The authors propose to associate PCOS and depression through the mediating factor of Future TP. While that logic is sound, the way it is applied in this paper is not intelligible. The authors seem to focus on the possibility that depression psychosomatically aids in the causation of PCOS while virtually ignoring the more likely scenario that people develop depressive symptoms as a response to a PCOS diagnosis. I can support publication if results are interpreted correctly/conservatively.

For example, in the abstract the authors propose to use TP in PCOS prevention programs. How and why would you do this? Are they assuming that an altered Future TP precedes PCOS and could be used to target women who may be at-risk? This has not been demonstrated and the more reasonable hypothesis is that Future TP becomes altered as a result of having PCOS. There is no conceivable causal relationship between Future TP and PCOS unless they establish some endocrinological basis for distorted TP. Vague references to inflammation and psychosomatic processes are insufficient to make such bold claims.

Furthermore, regardless of whether Future TP is a significant mediator of the relationship between depression and PCOS, the strength of these relationships is so small (0.11 and 0.09) such that, combined, there is less than a 1% influence. This is the same strength of influence as the direct, unmediated relationship between depression and PCOS. So what is TP adding here? At most, 1% extra explanation of a retrospective likelihood that a person has developed PCOS. Basically, all that this study has shown is that suffering PCOS leads to a change in Future TP, which is not surprising given that the Future TP is aligned with optimism and hope, things that are likely to be mediated by a chronic illness that could include infertility as a complication.

The authors claim that the hypothesis was supported (line 229), the hypothesis being that “there is a positive relationship between PCOS and depressive symptoms, and this relationship is explained by TP.”  While technically correct, they make no claim about how strong that relationship is and how it could be used to treat or prevent PCOS or depression. Perhaps one could look to secondarily prevent depression in those diagnosed with PCOS by improving their Future TP on the basis of these results but the suggestion that you might prevent PCOS through Future TP and unspecified psychosomatic processes is tenuous at best. Unless you can rule out the possibility that the women developed higher depression symptoms after being diagnosed with PCOS, then there are no conclusions to be drawn about prevention.

Other comments:

Line 58: 37% of people displaying symptoms of depression does not seem high by population standards. Is that at least one symptom? 37% would only be noteworthy compared to healthy controls. Either specify or consider removing.

Line 77: The treatment of TP is not technically accurate. It includes positive and negative attitudes to past and future but present hedonism and fatalism do not fit neatly into opposing positive and negative valences. Please rephrase

Line 79: What is a “temporal position”? TP is not a mental timeline and I’ve never heard this term used in reference to TP.

Line 261: Either provide a reference or please remove. “Often thought of” could be read as applying to the authors only.

Line 272: Given you found a non-significant direct relationship between depression and PCOS, why would you recommend the BDI? I would think, given the weakness of the evidence put forward, that no clinical recommendations should be made on the basis of these results. Further enquiry needed!!

Reviewer 2 Report

Thank you for the opportunity to read the text entitled "Time Perspective as a Mediator of Depressive Symptoms in Patients with Polycystic Ovary Syndrome".  Patients with PCOS who are significantly more often diagnosed with mood disorders and depression. Due to the prevalence of PCOS and the fact that this disease affects young women in the reproductive period, the work seems particularly necessary and interesting. The authors try to include a new element in the understanding of the problem of depressive disorders in PCOS patients.

The manuscript contains several shortcomings that should be addressed before publication in the journal Healthcare.

- please provide e-mail addresses for work, private addresses do not show the professional image of university employees

- please remove the markings (1), (2) etc. from the abstract, they are redundant elements

- line 85: in what role was the citation [28] introduced here. It is unclear to the reader. Please elaborate or remove this quote from the end of the line.

- add patient flow diagram. Describe, if possible, how many people and for what reasons were excluded from participation in the study

- materials and methods: divide this section into subsections: study group (with precise inclusion and exclusion criteria); biochemical research; psychological research, specifying the criteria for interpreting the results of answers given in the questionnaires. This layout will be clearer to the reader.

- provide a source containing guidelines for the ultrasound diagnosis of polycystic ovaries

- line 143: grammatical error

- lines 160, 229, 283: use an impersonal form of expression

- lines 232-234: how does the thought presented in these lines relate to the fact that in this work, the subjects significantly differ in age? Similarly: line 286: the groups differed in age, so the sentence is completely unclear.

- discussion. The first sentences of the discussion should show the most important observations of this paper. It's missing here. In this chapter, the authors talk extensively about the secondary results obtained in this project without touching on the most important topic. I would like the discussion to include more about TP and the possibility of practical application of this knowledge in working with PCOS patients.

Reviewer 3 Report

This is an interesting piece of work and the analyses seem appropriate. I only have one major concern and this has to do with the fact that, in your models, TP is the mediator of the association between depressive symptoms and PCOS. From my perspective, TP is more of a trait variable, whereas symptoms characterize the current state. In such a case, you might have expected tests of models where instead symptoms are a mediator of the TP-PCOS link. Of course cross-sectional mediation analyses cannot distinguish the merit of your versus the proposed, alternative, model, but I think that you should at least run such "reversed" models and report that they were done, and acknowledge it (probably also comment on it in the discussion). 
